

# PigLeg: prediction of swine phenotype using machine learning

Siroj Bakoev[1], Lyubov Getmantseva[1], Maria Kolosova[2], Olga Kostyunina[1], Duane R. Chartier[3] and Tatiana V. Tatarinova[4,5,6,7]

[1] L.K. Ernst Federal Science Center for Animal Husbandry, Moscow, Russia
[2] Don State Agrarian University, Persianovsky, Rostov Region, Russia
[3] ICAI, Culver City, CA, United States of America
[4] Department of Biology, University of La Verne, La Verne, CA, United States of America
[5] Institute for Information Transmission Problems, Russian Academy of Sciences, Moscow, Russia
[6] Vavilov Institute for General Genetics, Moscow, Russia
[7] Siberian Federal University, Krasnoyarsk, Russia

## ABSTRACT

Industrial pig farming is associated with negative technological pressure on the bodies of pigs. Leg weakness and lameness are the sources of significant economic loss in raising pigs. Therefore, it is important to identify the predictors of limb condition. This work presents assessments of the state of limbs using indicators of growth and meat characteristics of pigs based on machine learning algorithms. We have evaluated and compared the accuracy of prediction for nine ML classification algorithms (Random Forest, K-Nearest Neighbors, Artificial Neural Networks, C50Tree, Support Vector Machines, Naive Bayes, Generalized Linear Models, Boost, and Linear Discriminant Analysis) and have identified the Random Forest and K-Nearest Neighbors as the best-performing algorithms for predicting pig leg weakness using a small set of simple measurements that can be taken at an early stage of animal development. Measurements of Muscle Thickness, Back Fat amount, and Average Daily Gain were found to be significant predictors of the conformation of pig limbs. Our work demonstrates the utility and relative ease of using machine learning algorithms to assess the state of limbs in pigs based on growth rate and meat characteristics.

## INTRODUCTION

One of the main research tasks in animal husbandry is the discovery of the biological mechanisms influencing animal productivity and finding efficient ways of increasing it. Pork is the most widely consumed meat in the world. In addition to meat, many valuable products come from pigs: insulin, replacement human heart valves, suede for shoes and clothing, and gelatin for food and industry.

Intensive pig farming is associated with negative technological pressure on the development of pigs. Breeding for accelerated development and meatiness leads to a rearrangement of the metabolism in the animal's body, resulting in morphological and

Corresponding author
Siroj Bakoev, siroj1@yandex.ru

functional rearrangements of the internal organs, muscle, adipose, and bone tissues. Changes associated with the cartilage structure are called osteochondrosis (leg weakness). In industrial pig farming, the term "leg weakness" is used to describe the poor constitution of pig legs or the clinical condition associated with lameness or stiffness of movements. Such weakness results from abnormal changes in the cartilage joints and the development of epiphyseal plates, which are responsible for bone enlargement both in length and diameter (*Ekman & Carlson, 1998*). Weak epiphyseal plates can break, and the cartilage that covers the joint surface cracks. In the acute phase of the disease, bone fractures may occur near the epiphyseal plate. However, in most cases, the disease takes a chronic form, develops gradually, and manifests itself as incorrect shape and alignment of legs, as well as stiffness of the animal's gait. In this regard, the first step in diagnosing the disease is an exterior assessment of the legs and gait. Typically, pig legs are visually assessed by specially trained personnel using a point system (*Le et al., 2017*).

Rapid advances in next-generation sequencing (NGS) and high-density genotyping technologies allows identification of several quantitative trait loci (QTL) for pig lameness and leg weakness. Leg weakness is partially a heritable trait, with heritability estimates of leg ranging from low (0.07, *Aasmundstad et al., 2014*) to moderate (0.36, *Knauer et al., 2011*). Despite the agricultural importance of this trait, there has only been a limited number of GWAS for leg weakness. In addition, the trait may be complex and influenced by many factors, such as bone strength, muscle growth, fat accumulation, farming practices, animal activity level, and body weight gain. Therefore, one of the tasks of the present work was to identify these factors using modern statistical approaches.

Rapidly developing data mining approaches are of increasing interest because they provide for acquisition and analysis of information that results in predictive productivity indicators for animals (*Morota et al., 2018*; *Putz et al., 2018*; *Howard, 2019*). Machine learning (ML) approaches have been successfully used in animal husbandry for early prediction of the growth and quality of adult wool in Australian merino sheep (*Shahinfar & Kahn, 2018*), sheep carcass traits from early-life records (*Shahinfar, Kelman & Kahn, 2019*), and skin temperature of piglets (*Gorczyca et al., 2018*). Compared to other statistical approaches, ML is suitable for use even when there are many predictors, missing values, and abnormally distributed data, which is often the case with data obtained from commercial pig production.

In this work, we have evaluated the condition of pig legs by application of ML methods to growth and meat characteristics (see Fig. 1). We have compared common ML classification algorithms for predicting the state of the front and hind legs. This led to the identification of the most effective algorithm for predicting leg weakness using a small set of cost-effective and easily measurable sets of functions that can be used in the early period of animal rearing.

## MATERIALS AND METHODS

### Data sources

In pig farming, over the past few decades, a primary focus has been on improving meat quality, growth rate, and reproductive qualities of animals. The main parameters for

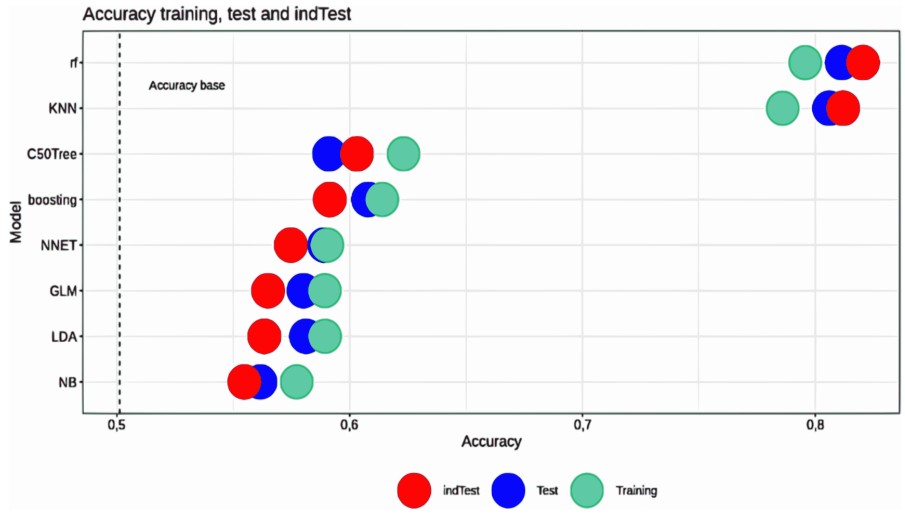

**Figure 1 Accuracy for training and testing sets for different ML approaches.** A graphical interpretation of the comparative analysis of the predicted value of all models shows that all models in the training set receive more accurate forecasts than in the test set. At the same time, the RF and KNN models provide high accuracy of prediction relative to other models. To determine the models that achieve the best results in solving the problem after the training procedures and their optimization, a comparative analysis was carried out. Obviously, the indicators obtained by validation are estimates of the ability of the model to predict new observations and these estimates have deviations.

**Table 1 Sample description.** The dataset contains 21,247 females and 3,337 males. 12,195 of Landrace and 12,389 of Large White breeds. Predictors: Average Daily Gain, Backfat Thickness, Muscle, Thickness, Birth Date, Breed, Sex. Dependent variables: scores for front and back legs.

| Variable | Min | 1st Qu. | Median | Mean | 3rd Qu. | Max |
|---|---|---|---|---|---|---|
| Average Daily Gain | 0.33 | 0.72 | 0.79 | 0.79 | 0.85 | 1.61 |
| Backfat Thickness | 4.30 | 10.90 | 12.90 | 13.25 | 15.20 | 35.60 |
| Muscle Thickness | 32.12 | 56.04 | 59.70 | 59.68 | 63.40 | 96.00 |
| Birth Date | 2012 | 2014 | 2015 | 2015 | 2016 | 2016 |
| **Scores** | | | | | | |
| Front Legs | 1.00 | 3.00 | 3.00 | 3.11 | 3.00 | 5.00 |
| Back Legs | 1.00 | 3.00 | 3.00 | 2.99 | 3.00 | 5.00 |

selection are the Average Daily Gain (ADG), Muscle Thickness (MT), and Bacon/Backfat Thickness (BF). In addition, we have investigated other factors that can affect the conformation of legs: breed, year of birth (Birth Date), and gender. The data were obtained from 24,584 pigs of breeds Landrace and Large White. Measurements were made *in vivo* using ultrasound scanners. ADG is measured in grams, MT and BF are measured in mm (see Table 1).

Front and Back legs were visually assessed using a point system from 1 to 5 (from bad to good). The assessment was performed by specially trained personnel. Points 1 and 2 were received by animals with visible leg defects, 3 points—average condition, 4 and 5—good and excellent, respectively. Preliminary data analysis showed the imbalance of the available

data. Imbalanced classes are a common problem in machine learning classification where there is a disproportionate ratio of observations in each class. Since most ML algorithms work best when the number of samples in each class are about equal, a balancing procedure was applied. After the preliminary analysis, the year of birth and gender were excluded as predictors of the least importance.

## Methods

The algorithms used for this work were selected from a wide range of probabilistic and non-probabilistic methods in order to cover the entire spectrum of existing tools. In addition to the algorithms presented in our work, we tested other methods, for example, the XGboost algorithm, which is commonly used for solving classification problems, but in our case, it only gave mediocre results. In agricultural mathematics, from the beginning of the 20th century to this day, the statistical methods of Fisher and Wright have been widely used. With the successful use of ML algorithms in various fields of human activity, it was inevitable they have appeared in agricultural problem-solving.

Classification models were constructed and analyzed using the following ML methods: Random Forest (RF) (*Rennie et al., 2003*), K-Nearest Neighbors (K-NN) (*Walker & Duncan, 1967*; *Breiman, 1998*), artificial neural networks (Neural Networks) (*Fisher, 1936*), C50Tree, Support Vector Machines (SVM) (*R Core Team, 2013*), Naive Bayes (NB) (*Kuhn, 2008*), GLM (*Torgo, 2011*), Boost (*Shitikov & Mastitsky, 2017*) and Linear Discriminant Analysis (LDA) (*Vapnik & Chervonenkis, 1974*). All calculations and simulations were performed in R (version 3.6.1, *Ripley & Venables, 2011*) using the caret packages (*Friedman, 1999*), DMwR (*Mason et al., 2000*). Leg scores were used as the response variables.

## K-Nearest Neighbors (K-NN)

The K-NN classifier is based on the compactness hypothesis, which assumes that a test object will have the same class label as the training objects in the local area of its immediate environment. When the value of K is one, the analyzed object is assigned to a certain class depending on information about its single nearest neighbor. When $K > 1$, every object is assigned to the prevailing class of nearest neighbors. Any clustering algorithm can be considered effective if the *compact hypothesis* is satisfied, meaning that there exists a partition of objects into groups that the distances between objects from the same group (intra-cluster distances) will be less than a certain value $\varepsilon > 0$, and the distance between objects from different groups (cross-cluster distances) is more than $\varepsilon$ (*Jørgensen & Andersen, 2000*).

## Linear Discriminant Analysis (LDA)

LDA is a multidimensional analysis section that allows one to evaluate differences between two or more groups of objects using several variables. It is a generalization of Fisher's linear discriminant, a method used in machine learning to find a linear combination of features which characterizes or separates two or more classes of objects or events. The resulting combination can be used as a linear classifier or, more often, to reduce the dimension before subsequent classification. LDA is closely related to the analysis of variance (ANOVA) procedure. The LDA implements two closely related statistical procedures:

1. Interpretation of group differences, needing to answer the question: how a well-used set of variables can form a dividing surface for objects of the training sample and which of these variables are the most informative.
2. Classification, i.e., prediction of the value of the grouping factor for the examined group of observations.

## The support vector machines (SVM)

SVM, previously called the "generalized portrait" algorithm, was developed by Soviet mathematicians Vapnik and Chervonenkis (*Nakano, Brennan & Aherne, 1987*) and has since gained widespread popularity. The main idea of the classifier on support vectors is to build a separating surface using only a small subset of points lying in the zone critical for separation, while the rest of the correctly classified observations of the training sample outside of this zone are ignored (more precisely, they are a "reservoir" for an optimization algorithm). If there are two classes of observations and a linear form of the boundary between the classes is assumed, then two cases are possible. The first of them relates to the possibility of perfect data separation with the help of some hyperplane. Since there can be many such hyperplanes, the dividing surface is optimal, which is as far as possible from the training points, i.e., having a maximum gap $M$ (margin).

## Naive Bayes classifier (NB)

Naive Bayes classifiers are a family of simple probabilistic ML classifiers based on the application of Bayes theorem. Making the "naive" assumption that all the signs describing the classified objects are completely equal and are not related to each other, then the probability of an object to belong to a given class given its observed features, $P(class|features)$, is calculated using the Bayes formula from known distributions $P(features|class)$. The NB assigns the objects to refer to the class that has the greatest probability.

## Neural networks

Neural network models that were born in the process of developing the concept of artificial intelligence have two completely transparent analogies—the biological neural system of the brain and the computer network. Their main paradigm is that the solution in the network is formed by many simple neuron-like elements that form a graph with weighted synaptic (informational) connections that work together and purposefully to obtain a common result. To train artificial neural networks in the R environment, the *nnet* package (*Lundeheim, 1987*) was used; it provides flexible functionality for constructing classification models based on a multilayer perceptron.

## GLM

Logistic regression is commonly used as a binary classifier for alternate response samples. However, this method can also be generalized to the case with several classes. Nominal or ordinal variables can be used as the simulated response Y, and in both cases, a multidimensional binomial distribution is assumed. Simply put, linear regression should be used to predict a quantitative (i.e., numerical) response variable, and logical regression

should be used to predict a qualitative (i.e., categorical) response variable. Both linear regression and logistic regression are types of generalized linear models (GLM).

## Gradient boosting

One of the methods for improving predictions is boosting, which is an iterative process of sequentially constructing private models. Each new model is trained using the information on errors made at the previous stage, and the resulting function is a linear combination of all, considering the minimization of any penalty function. Like bagging, boosting is a general approach that can be applied to many statistical classification methods. The idea of increasing the gradient arose as a result of Leo Braiman's observations that increasing the gradient can be interpreted as an optimization algorithm on an appropriate cost function. Several algorithms for increasing the gradient of direct regression were developed (*Van der Wal et al., 1980*). The *Draper, Rothschild & Christian (1992)* approach optimizes the cost function with respect to the functional space by iteratively choosing a function, indicating the direction of the negative gradient.

## The C 50Tree method

This method is based on the application of a strategy of dividing data into smaller and smaller parts to identify patterns that can ultimately be used for forecasting. The model itself includes many logical decisions, with decision nodes. They are divided into branches that indicate the choice of solution. The tree ends with leaf nodes (also called terminal nodes), which indicates the result of a combination of decisions. The data to be classified begins at the root node, where the ripple is transmitted to them, and various decisions in the tree, in accordance with the values of the predictors, depending on their influence on the response variable.

## Random forest

Random Forest is a controlled learning method in which the target class is *a priori* known, and a model is built (classification or regression) to predict future responses. Several hundred decision trees are built for training bootstrap samples. However, at each iteration of the tree construction, randomly selected $m$ from $p$ predictors to be considered, and the partition can be performed on only one of these $m$ variables. The meaning of this procedure, which turned out to be very effective for improving the quality of the obtained solutions, is that with the probability $(p-m)/p$ some potentially dominant predictor that seeks to enter every tree is blocked. By blocking dominants, other predictors will get their chance, and tree variation will increase.

## Data preparation

The number of observations for training models allows one to achieve high predictive effectiveness. The data includes both continuous and high-quality variables, which allows facile problem-solving. The response variable (target variable) was a leg score, which varies from 1 to 5. For practical reasons, the values were adjusted and divided into two bins: scores [1:2]—animals with "bad" legs (Q1) and scores [3:5]—animals with "good legs" (Q2). Accuracy was calculated as the proportion of correct predictions of the algorithm, precision,

recall, and the F1 score (a harmonic mean of precision and recall). The data points were assigned to the bins (2708 (4930) for Q1; 21876 (19654) for Q2), corresponding to 11% (20%) and 89% (80%) of measurements for the two breeds. The imbalance of the data classes (a large difference between the numbers of samples in different bins) can negatively affect both the learning and prediction phases of the approach. If the unbalance ratio is high, the decision function favors the "majority" class, where the largest number of samples is located. Not all ML models are affected by unbalanced classes and most probabilistic models are weakly dependent on them. However, problems arise when non-probabilistic classifiers are used. For example, in logistic regression, neural networks, as well as in SVM algorithms, class balance strongly affects their parameters. In the decision trees, random forest, and gradient boosting approaches, class imbalances affect the measures of leaf impurity. To solve the problem of class imbalance, the oversampling method was used. The advantage of using this method is that it does not lead to information loss. Therefore, the leg score data for the bins were balanced using the ROSE package.

## Data analysis

Before choosing the most important predictors and training the prognostic model, a descriptive study of variables was conducted. This process allows for a better understanding of what information each variable contains, as well as to identify possible errors. The procedure for collecting information on farms is determined by the human factor as are other production features of the industry. It is not always possible to enforce measurement collection protocol compliance and obtain data in its entirety; therefore, missing values occur in our dataset. To fill in the missing values, we used the *preProcess* function from the *caret* package in R (classification and regression training, http://topepo.github.io/caret/index.html, *bagimput* method). This method constructs a "bagging" model for each of the available variables based on regression trees, using all other variables as predictors; it requires significant computation time, especially when working with large data sets (*Shitikov & Mastitsky, 2017*).

Studying the distribution of the response variable relative to quantitative (Muscle Thickness, Back Fat, Average Daily Gain) and qualitative (Breed) variables is an important exercise. Analysis of quantitative variables showed a pronounced asymmetric distribution of some predictors (Back Fat). The calculation of correlations between continuous predictors indicates that they do not contain redundant information (Fig. 2).

For a predictive model to be useful, it must have a success rate higher than expected by chance or at a certain base level. In classification problems, the base level is the level obtained if all observations are assigned to the majority class. In our case, since 89% (80%) of the animals have healthy front (hind) legs, then the expected success rate is 89% (80%) for unbalanced and 50% for balanced data participating in the training set. Our goal to design predictive models that have a better success rate than the expected one. Since the aim of the study is to assess the state (conformation) of legs by means of selected predictors (e.g., growth and meat quality), we are interested in the proportion of the animals with healthy legs (correspondence with other leg conformation classes is less important). By analyzing the data in this way, one can begin to extract ideas about which variables are

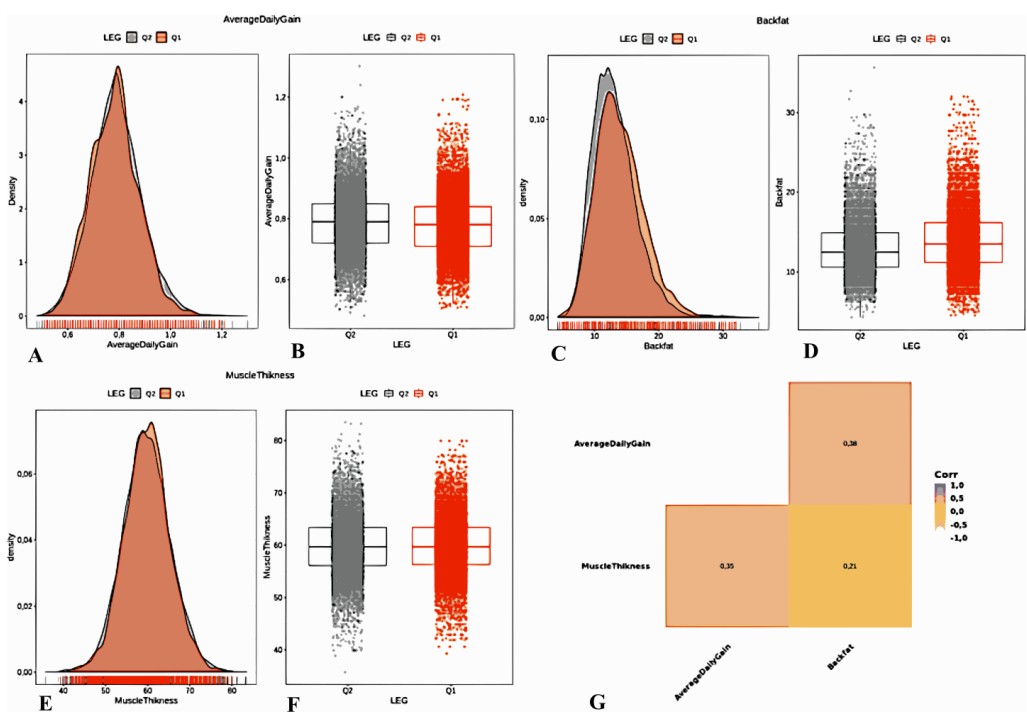

**Figure 2** **Analysis of the collected measurements.** Variables (Average Daily Gain (A, B), Back Fat (C, D), and Muscle Thickness (E, F)). Back Fat (C) has an asymmetric distribution. Concordance analysis of predictors (G) shows a moderate correlation between the three parameters. This indicates that they do not contain redundant information.

most associated with "good" legs. To study the importance of predictors, we also used the Random Forest package. All studies algorithms have identified that the most important predictors are Muscle Thickness, Back Fat, Average Daily Gain, while the predictor Breed is not significant (Fig. 3).

## Model training

Figure 2 shows that the measurements follow bell-shaped distributions. Therefore, a standardization of the data was carried out by subtracting the mean and dividing each predictor by its standard deviation, so the data obeys the standard normal distribution.

Machine learning algorithms were trained and tested based on the following structure for all three features of interest in this study. A random 10% of the data was excluded from the complete data set for the final assessment, that we designate for an independent trial. The independent test dataset was not used to build a model, it was only used to test the models (results are shown in Fig. 1 as the *indTest*). The remaining 90% of samples were randomly divided into 70% for training and 30% for testing; the process was repeated 100 times. In every 100 training iterations, hyperparameters were selected using a search within the 10-fold cross-validation structure on a random 70% subset of the training set. The selected hyperparameters were used to train each ML model on a training set and were

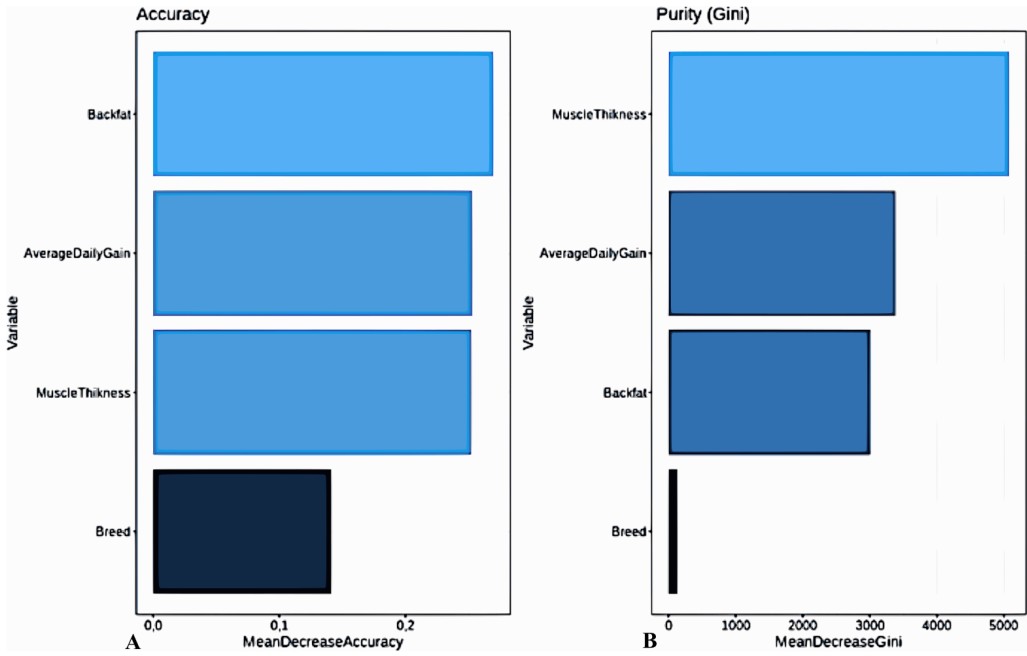

**Figure 3** **Relative importance of leg weakness predictors as assessed by Accuracy (A) and Gini (B).** Since the aim of the study is to assess the state (conformation) of legs by means of selected predictors (growth and meat quality), each variable is analyzed with respect to the variable Q2 = "good". By analyzing the data in this way, one can begin to extract ideas about which variables are most associated with "good" legs. Alternatively, to study the importance of predictors, we use the Random Forest package. All studies algorithms have identified that the most important predictors are Muscle Thickness, Back Fat, Average Daily Gain, while the predictor Breed is not significant.

tested on a test set in each iteration. All processes were implemented in R. The performance of the final model has been evaluated on the test and on the *indTest* sets.

## Performance assessment

Model fit and ranking between models is assessed using several scores that can be computed from the number of true positive (TP), true negative (TN), false positive (FP), and false-negative (FN) predictions. These numbers can be applied per class or be aggregated for the entire dataset.

Accuracy measures a fraction of correct predictions as is usually represented as a percentage.

$$\text{Accuracy} = \frac{\text{TP} + \text{TN}}{\text{TP} + \text{TN} + \text{FP} + \text{FN}}.$$

Error rate measures a fraction of incorrectly classified samples.

$$\text{Error} = \frac{\text{FP} + \text{FN}}{\text{TP} + \text{TN} + \text{FP} + \text{FN}} = 1 - \text{Accuracy}.$$

Specificity (Precision) is the fraction of correct predictions.

$$\text{Precision} = \text{Specificity} = \frac{\text{TP}}{\text{TP} + \text{FP}}$$

Sensitivity (Recall) measures a fraction of correct predictions per the true number of samples.

$$Recall = Sensitivity = \frac{TP}{TP+FN}.$$

The $F$-Measure ($F1$) is a goodness of fit assessment for a classification analysis that balances precision and recall, ranging between 0 and 1.

$$F1 = 2\frac{Precision \times Recall}{Precision + Recall}.$$

Cohen's Kappa is a common measure to calculate agreement between the classification of qualitative observations. Let $p_e$ be the expected number of agreements and $p_o$ be the observed number of agreements.

$$\kappa = \frac{p_o - p_e}{1 - p_e}.$$

## RESULTS

The ML approach was able to predict the state of the front and hind legs. The Random Forest method surpassed all other learning algorithms in all tested scenarios. The KNN approach was a close runner-up (Table 2). The superiority of RF and KNN is due to a lower value of the variance of forecasting indicators as compared to SVM and NB approaches. SVM and NB were among the least effective forecasting methods in this study, providing the lowest correlation and the largest forecasting errors. A graphical interpretation (Fig. 1) of the comparative analysis of the predicted value of all models shows that all models in the training set receive more accurate forecasts than in the test set. In order to address the overfitting issue, we have replaced the ''KNN'' method by the ''KKNN'' method. KKNN method is more flexible and permits selection of the measure of similarity, the shape of the kernel function, as well as the estimation of the optimal value of the parameter k using the cross-validation approach. In the case of the Random Forest algorithm, varying parameters of cross-validation, selection of the optimal cut-off value, the accuracy can be improved. Using an independent subset of data, we have demonstrated that the overfitting issue was resolved.

Both RF and KNN models provide higher accuracy of prediction compared to other models. To determine what models achieved the best results in solving the problem after the training procedures and their optimization, a comparative analysis was carried out. Obviously, the indicators obtained by validation are estimates of the ability of the model to predict new observations and these estimates have deviations. A comparison was made between all models with the non-parametric Friedman test and a pairwise comparison of all models, the results of which are summarized in Table 3. The best predictive capabilities in the dataset were shown by the Random Forest approach. In addition, it must be noted that such signs as Muscle Thickness, Back Fat, Average Daily Gain can act as predictors of leg weakness. Information on breed and gender was not significant for assessing the status of legs.

**Table 2 Comparison between the models using the testing dataset.** ML models were able to predict the state of the fore and hind legs. RF surpassed all other learning algorithms in all respects and scenarios. In some cases, RF did not have significant superiority over KNN. Accordingly, KNN was the second most efficient algorithm among all the characteristics and scenarios.

| Model | Accuracy | Kappa | $P$-value $\kappa$ | Sensitivity | Specificity |
|---|---|---|---|---|---|
| RF | 0.8846 | 0.7693 | <2.2e−16 | 0.8232 | 0.9463 |
| KNN | 0.8754 | 0.7509 | 3.238e−16 | 0.8013 | 0.9499 |
| C50Tree | 0.6469 | 0.294 | 0.001603 | 0.5746 | 0.7195 |
| Boost | 0.6035 | 0.207 | 0.09968 | 0.5995 | 0.6075 |
| NNET | 0.5667 | 0.1335 | 0.01852 | 0.5619 | 0.5716 |
| LDA | 0.563 | 0.1258 | 2.343e−05 | 0.5986 | 0.5272 |
| GLM | 0.5624 | 0.1246 | 5.043e−05 | 0.5971 | 0.5275 |
| SVM | 0.5603 | 0.1202 | 3.248e−05 | 0.653 | 0.4671 |
| NB | 0.5411 | 0.0816 | <2.2e−16 | 0.6984 | 0.3832 |

**Table 3 Search results among all the models of non-parametric tests of Friedman and paired comparison of all models.** A comparison was made between all models with the non-parametric Friedman test and a pairwise comparison of all models. The best predictive capabilities in the dataset were shown by the Random Forest approach. In addition, it must be noted that such signs as Muscle Thickness, Back Fat, Average Daily Gain can act as predictors of leg weakness. Information on breed and gender were not significant for assessment the status of legs.

| Model A | Model B | $p$-value | Model A | Model B | $p$-value |
|---|---|---|---|---|---|
| boosting | arbol | 4.37E−02 | NB | logistic | 2.17E−08 |
| KNN | arbol | 2.17E−08 | NET | arbol | 2.17E−08 |
| KNN | boosting | 2.17E−08 | NET | boosting | 2.17E−08 |
| LDA | arbol | 2.17E−08 | NET | KNN | 2.17E−08 |
| LDA | boosting | 2.17E−08 | NET | LDA | 1.31E−07 |
| LDA | KNN | 2.17E−08 | NET | logistic | 1.31E−07 |
| logistic | arbol | 2.17E−08 | NET | NB | 2.17E−08 |
| logistic | boosting | 2.17E−08 | rf | arbol | 2.17E−08 |
| logistic | KNN | 2.17E−08 | rf | boosting | 2.17E−08 |
| logistic | LDA | 9.30E−02 | rf | KNN | 2.17E−08 |
| NB | arbol | 2.17E−08 | rf | LDA | 2.17E−08 |
| NB | boosting | 2.17E−08 | rf | logistic | 2.17E−08 |
| NB | KNN | 2.17E−08 | rf | NB | 2.17E−08 |
| NB | LDA | 2.17E−08 | rf | NET | 2.17E−08 |

Friedman rank sum test

Friedman chi-squared = 286.85, $df = 6$, $p$-value < 2.2e−16

## DISCUSSION

The increase in the prevalence of leg weakness in pigs in the middle of the 20th century coincided with a surge of targeted breeding work to increase the growth rate of animals. This was mainly due to economic pressure and the need to shorten the period from birth to slaughter. Since in wild boars, requiring about two years to reach maturity, osteochondrosis is not observed, it was proposed that there was a relationship between the

growth qualities and weakness of the legs. Several large population studies have shown a positive correlation between these traits (*Ekman & Carlson, 1998*; *Breiman, 2001*; *Cover & Hart, 1967*). Lundeheim (*Lantz, 2015*) noted that pigs with clinical signs of leg weakness grew faster in the early stages of life than pigs without these signs, but by the time of slaughter, their growth had become slower. He suggested that the unfavorable relationship between fatness and growth rate is balanced by discomfort due to the emerging clinical signs of leg weakness, leading to reduced feed intake. Van der Wal et al. (*Ripley & Hjort, 1996*) discovered a significant correlation between the length of the carcass and the weight of the ham with the degree of damage to the proximal and distal parts of the femur—osteochondrosis. The relationship between the state of the legs and indicators of meat productivity of pigs was confirmed by several studies conducted on pigs of various breeds. A study by Draper et al. (*Cortes & Vapnik, 1995*) showed that Duroc pigs with low foreleg scores had greater muscle length and mass. Draper et al. examined the thickness of fat, the length of the body and the yield of meat but found no significant differences related to the condition of the legs. In another study, the emphasis was placed on studying the relationship between the legs and meat qualities of large white pigs. The results showed that pigs with leg problems were usually heavier and with more back fat compared to healthy pigs. These observations agree with the results obtained by our machine learning approach. Therefore, we have demonstrated that machine learning can be successfully used to evaluate the growth performance and meat characteristics of pigs.

## CONCLUSIONS

Leg weakness is a source of significant economic loss in pig production, therefore, the search for predictors of leg condition is of great interest and potential value. Machine learning is a relatively new paradigm in computational biology. The problem of processing and comprehending a huge data stream poses a challenge for researchers to develop new computational methodologies. In our opinion and experience, ML algorithms are a good alternative to parametric models to solve many problems in biology. ML focuses on algorithmically constructed models with optimal forecasting as their ultimate goal. One of the important additions to the accurate forecasting of ML is the ability to obtain data for training from empirical observations and use ML to train algorithms for recognizing phenomena that may be overlooked. Our comparison of various machine learning algorithms proved that growth rate and meat parameters were effective predictors of the condition of pig legs. PigLeg provides a powerful tool to assess the health of the animals. The best predictive performance was achieved by the Random Forest approach.

**Abbreviations**

| | |
|---|---|
| **NGS** | Next-Generation Sequencing |
| **QTL** | Quantitative Trait Locus |
| **GWAS** | Genome-Wide Association Studies |
| **ML** | Machine Learning |
| **ADG** | Average Daily Gain |
| **MT** | Muscle thickness |

| **BF** | Back Fat |
| **RF** | Random Forest |
| **KNN** | K-Nearest Neighbors |
| **NN** | Neural Networks |
| **SVM** | Support Vector Machines |
| **NB** | Naïve Bayes |
| **GLM** | Generalized Linear Models |
| **LDA** | Linear Discriminant Analysis |

### Funding
Funding was provided by the Russian Foundation for Basic Research 19-016-00068 A. The funders had no role in study design, data collection and analysis, decision to publish, or preparation of the manuscript.

### Grant Disclosures
The following grant information was disclosed by the authors:
Russian Foundation for Basic Research: 19-016-00068 A.

### Competing Interests
Tatiana Tatarinova is an Academic Editor for PeerJ. Duane Chartier is the founder of ICAI, Inc.

### Author Contributions
- Siroj Bakoev and Tatiana V. Tatarinova analyzed the data, prepared figures and/or tables, authored or reviewed drafts of the paper, and approved the final draft.
- Lyubov Getmantseva, Maria Kolosova and Olga Kostyunina conceived and designed the experiments, performed the experiments, prepared figures and/or tables, and approved the final draft.
- Duane R. Chartier conceived and designed the experiments, prepared figures and/or tables, authored or reviewed drafts of the paper, and approved the final draft.

### Data Availability
Software is available at: https://github.com/Khalimat/PigLeg.
The raw data and software are available at: http://www.compubioverne.group/data-and-software/.

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
