# Peer review of "PigLeg: prediction of swine phenotype using machine learning"

_PeerJ, doi:10.7717/peerj.8764_

## Round 0.1 · original submission · Minor Revisions

Two specialists in the field evaluated this submission. They see merits in the manuscript and suggest minor revisions. Please ensure that the English language in this submission meets our standards: uses clear and unambiguous text, is grammatically correct, and conforms to professional standards of courtesy and expression. Considering the evaluation carried out by these reviewers, I recommend minor revision in this paper.

Reviewer 1 ·

Basic reporting

This manuscript, to the reviewer, is well-organized and easy to read. The use of English is acceptable. Literature review is also enough and relevant.

Experimental design

This study fits the journal well. The research question is well-defined. The data collection process needs more explanations.

Validity of the findings

The proposed model might be overfitting. This should be solved.

Additional comments

This study is interesting for adopting machine learning approaches to predict leg weakness and lameness of pigs. Their findings can have important practical implications. Several suggestions are provided for the authors.

1.‘Materials and Methods’ section: How predictors are selected for building the model should be explained.
2.‘Materials and Methods’ section: It is suggested to present the operational definitions and measurement scale of predictors used in this study. Further, how these predictors are collected should be delineated as well. If data collection involves human judgment, what are the qualifications of those raters? And also how to ensure the reliability of those collected data?
3.‘Materials and Methods’ section: Is there any particular considerations for choosing these nine machine learning algorithms?
4.‘Data Preparation’ section: Regarding class imbalance issue, it is suggested to explicitly express what method (under- or over-sampling) is used to solve the issue instead of saying ‘…using the ROSE package…’
5.‘Model training’ section: This study randomly chose 10% of the sample for the final assessment, nonetheless no assessment results was demonstrated in this study.
6.‘Results’ section: The proposed model is suffered from ‘overfitting’ because the prediction accuracy of train data is higher than that of test data (lines 238-239). This issue should be solved or the proposed model is unable to predict new data accurately.

Reviewer 2 ·

Basic reporting

See below

Experimental design

See below

Validity of the findings

See below

Additional comments

The paper can be accepted but it needs to be corrected and updated. The comments are:

a) There are many grammatical mistakes such as:

Pork is the most widely consumed meats in the world

b) Description of the data balancing techniques used is not given

c) These also exist some classifiers whose performances are least affected by class imbalance. Comments on the that is also required.

d) The reason of using a method for filling the missing values needs to be quoted. And also it needs to mentioned that who effective their approach was?

e) The significance of Kappa value (Table 1), Friedman rank (Table 2) along with the other metrics needs to be mentioned. Also it needs to be mentioned the relevance of these tests w.r.t. these results.

f) The description of the data used needs more details

g) Figure 1 is confusing w.r.t. RF and KNN classifiers

h) Figure 2 and figure 3 are not readable, quality of these figures need to be improved

i) More emphasis can be given on the proposed methodology and the conclusions of this work

---

## Round 0.2 · accepted · Accept

The revised manuscript improved a great deal and can be accepted as it is.

Reviewer 1 ·

Basic reporting

No comment.

Experimental design

No comment.

Validity of the findings

No comment.

Additional comments

The authors have addressed reviewers' comments.